# Electronic Patient-Reported Outcome Measures in Burn Scar Rehabilitation: A Guide to Implementation and Evaluation

**Jill Meirte** [1,2,*] and **Zephanie Tyack** [3,4,†]

1. Department of Rehabilitation Sciences and Physiotherapy REVAKI-MOVANT, Faculty of Medicine and Health Sciences, University of Antwerp, 2610 Wilrijk, Belgium
2. Oscare, Organisation for Burns, Scar After-Care and Research, 2170 Antwerp, Belgium
3. UQ Child Health Research Centre, The University of Queensland, 62 Graham St, South Brisbane, QLD 4101, Australia; z.tyack@qut.edu.au
4. Australian Centre for Health Services Innovation (AusHSI), School of Public Health and Social Work and Centre for Healthcare Transformation, Queensland University of Technology, 60 Musk Avenue, Kelvin Grove, QLD 4059, Australia
* Correspondence: jill.meirte@uantwerpen.be
† These authors contributed equally to this work.

**Abstract:** In burn scar rehabilitation, electronic patient-reported outcome measures (ePROMs) are increasingly being used in research and clinical settings as part of patient- and family-centred care. These measures can identify patients' needs and monitor the therapeutic progress of both adults and children. The feedback of information from ePROMS to clinicians treating patients with scarring and psychosocial issues may have therapeutic benefits. However, testing the effectiveness of ePROMs used in the routine clinical care of patients with burn scarring is in its infancy, and one of the greatest challenges remains the implementation of ePROMs in real-world clinical settings. The aim of this paper is to provide a guide for clinicians and researchers involved in burn scar rehabilitation to assist in implementing ePROMs in clinical settings. The guide outlines strategies, processes, and considerations for ePROM implementation and the accompanying resources. Two real-world case studies of ePROM implementation are presented in burn scar clinics in Belgium and Australia. Additionally, ten recommendations for the implementation of ePROMs are provided based on research evidence and the lessons learned by the authors. The information provided should pave the way forward for using and testing these ePROMs in research and practice.

**Keywords:** implementation science; electronic patient-reported outcomes; burn; scar

## 1. Introduction

Increased attention is being given to patient- and family-centred care in which the voices of patients and families are heard and where they play an active role not only in treatment but also in research as research partners [1,2]. Patient-reported outcome measures (PROMs) are being used in research and clinical settings as part of patient-and family-centred care to identify patients' needs and monitor the therapeutic progress of both adults [3] and children [4]. The COVID-19 pandemic has helped to highlight the value of PROMs as part of telehealth interventions [5]. In the words of Nick Black—a leader in PROM research, "Patient Reported Outcome Measures could help transform healthcare" [6].

PROMs are questionnaires completed by patients which measure their perceived health status, symptoms, functional status, or (health-related) quality of life [7]. These measures may be either generic (not burn-specific) or condition-specific (specific to the burn injury or scarring) and may be used for multiple purposes, including screening, diagnosis, prognosis, follow-up, and planning. Routine clinical assessment of patients' treatment, scarring, and psychosocial functioning [8] can be achieved using PROMs, with potential therapeutic benefits [9,10]. To achieve this, PROMs are ideally chosen to capture areas that are most meaningful to the patients and their families.

Testing the effectiveness and implementation of the patient-reported outcome measures used in the routine clinical care of scars is in its infancy. However, rigorous studies across a range of areas in health care have identified some positive findings. Studies examining the acceptability and feasibility of these measures in clinical practice have indicated generally positive findings for patient–clinician communication [5] but mixed findings for integration within the existing workflows and an unclear impact on patient outcome and clinical parameters [11–14]. A recent Cochrane systematic review found that PROMs feedback probably improves quality of life and increases patient–physician communication, diagnosis and notation, and disease control with moderate certainty [15]. However, further rigorous research is needed to examine the effectiveness and implementation of using ePROMs in burn scar care. Using implementation science could assist in reducing the evidence-to-clinical-practice gap [16].

As in other fields of care, burn scar care is faced with a clear shift towards digital health and e-health. The development of the internet, mobile information-sharing technologies, and the widespread use of computer tablets and smartphones has led to the emergence of electronic PROMs (ePROMs). Generic, burn- and scar-specific PROMs that were originally developed using pen and paper have been transferred into ePROMs and electronic care pathways in clinical practice.

For centres where paper-based PROMs have already been implemented, potential benefits of moving to ePROMs have been reported. These include greater alignment with patient preferences and acceptability and lower costs, and similar or faster completion times for ePROMs. Higher data quality and response rates, and better symptoms' management and patient–clinician communication, have also been reported [17]. However, one of the greatest challenges remains the implementation in real-world clinical settings.

The aim of this paper is to provide a guide for clinicians and researchers involved in burn scar rehabilitation in order to assist with the implementation of ePROMs in the clinical setting. This implementation may be conducted as part of a research, clinical, or quality assurance initiatives; and targets healthcare clinicians as well as policy makers and researchers working in acute hospital, subacute, or after-care settings delivering burn care. It is intended that the guide and evidence base presented will be refined and updated as it is tested and applied in practice.

## 2. Organisation of the Paper and Context behind Recommendations

Our paper is organised into three sections. Firstly, we present the processes and strategies to support our guide to implementing ePROMs by addressing issues related to technology platforms and systems, privacy and confidentiality, different modes of presentation (for example, paper and electronic), PROM psychometrics, and the acceptability and feasibility of ePROM implementation in practice. Secondly, we present two case studies illustrating these processes. Finally, we present our top 10 recommendations for ePROM implementation based on the lessons we have learned in delivering these measures as part of burn scar care.

Systematic reviews, best-practice guidelines, and original studies in burns and scars have been used to formulate the guide. Systematic reviews on the barriers to and facilitators of the implementation of ePROMs have informed the processes and strategies outlined [17,18]. Best-practice guidelines have informed guidance regarding the selection of outcome measures (e.g., ISOQOL [19,20] and COSMIN guides [21]), paediatric administration [22], and moving from paper-based to electronically administered PROMs [23]. These guidelines have been detailed in Appendix A.

The conceptual frameworks drawn upon include those developed in the field of burn scar management, quality-of-life and well-being research—the ultimate goal of burn scar care, and implementation science. These frameworks have informed the selection of the outcomes, study processes, and evaluation processes underlying the case studies. The experience of the authors in developing, validating, and implementing scar-related ePROMs and digital scar pathways have also contributed to the guide. For example, this experience

has highlighted the importance of obtaining feedback from patients with burn scars and their families regarding the timing, length, content of PROMs, and IT barriers early in the implementation process. Questions commonly asked by clinicians and researchers seeking to commence the implementation of ePROMs in burn settings, as well as broader hospital settings, have also been covered in our guide. These questions have arisen as part of three local and international special interest groups that one of the authors has regularly attended over the last 2 years (in implementation science, quality of life research, and implementing PROMs and PREMs), and in the clinical settings where the authors conduct their research.

## 3. Body Section

### 3.1. Processes and Strategies for Implementing ePROMs

3.1.1. Preparing for Implementation

- Population

Whether or not you work in the burn scar rehabilitation setting or beyond, considering the characteristics of the target population, including age and technology literacy, are important. There are unique challenges when implementing child and adolescent PROMs. It has been established that reliability and validity of self-report is questionable below the age of 8 years. For the ages of 8 to 11 years, the reliability and validity of the child report improves, and between 12 and 18 years, self-report is preferred [22]. Research has shown that for children ePROM administration using the internet can be feasible, reliable, and valid [24–26]. Screen-based modes of administration can help children stay focused and engaged and improve the quality of the self-reported data, while minimizing missing data [17]. Overall, younger patients prefer ePROMs over paper formats, as they may be more familiar with the use of the internet and have access to necessary technological resources [17,27–29].

It should be noted that ePROM research with children requires careful attention to all aspects of the data collection process. Researchers need to use age-appropriate language for explaining the study purpose and procedures to children. Before children begin completing ePROMs, interviewers or other study staff members should inform them of what will be required of them, the purpose of the questions that will be asked, the intended use of the data, the confidentiality procedures, and what to do if they become uncomfortable or want to stop participating. While some children may not entirely comprehend the study details, researchers should make an effort to give children a general understanding and ensure that they feel comfortable [22].

People who are computer illiterate, older, or have no access to infrastructure could potentially be disadvantaged when ePROMs are implemented [29–33]. Be aware of the digital divide: some patients may be less willing or even unable to complete ePROMs without assistance due to computer illiteracy or to having no access to the internet or technological devices [17]. Checking the functional ability, mental health, and cognitive capacity of the target population [23] is advised as challenges in these areas may interfere with timely completion. Our experience indicates that in some cases the burden of completion may be too high and alternate methods of eliciting information are needed, as outlined in the section addressing equity.

Knowledge regarding the technology literacy of the target population and patients is important (i.e., experience using digital technology and computers, access to a computer/tablet/smartphone, or internet access at home). Limited experience by patients in using technology may affect their satisfaction, preference, and willingness to use electronic formats [17]. An educational session may not be necessary for everyone; however, it may be indispensable when working with children, people working with tablets for the first time, and computer illiterate and older people to provide instructions and determine whether assistance is required [25,32,34,35].

- Context and setting

The ability for ePROMs to be completed in the clinical setting, research area, or home setting should be checked prior to implementation. This includes checking internet connectivity in all the locations where ePROMs will be administered and being aware that medical equipment may interfere with the signal.

- Resources

Appendix A provides resources for researchers and clinicians covering the relevant frameworks, guidelines, reports, and papers to consider for different aspects of ePROM implementation in burn scar rehabilitation.

Other resources to consider are the specific equipment that is necessary for starting ePROM implementation. Be aware of one-time investments and costs at the start. Although overall the implementation of ePROMs can be more economic than the usual care [27,36], an initial one-time large investment may be needed. This may involve the purchasing of tablets or wireless printers (if paper printing is necessary), the hiring of computer programs or infrastructure for the collection of data, and the costs for online and technology support. When using a web-based platform, access to the internet is needed and may require procuring cellular (3G/4G) internet.

- Mode of administration

The implementation of ePROMs has been reported to facilitate improved data quality, completion times equal to or faster than paper-based PROMs, lower administration times, and better clinical decision making and symptom management [17]. However, the consideration of special populations is required when implementing web-based questionnaires with, for example, patients with low economic resources and patients unfamiliar with internet use [27].

For the transition of paper PROMs to ePROMs, the International Society for Pharmacoeconomics and Outcomes Research (ISPOR) offers a clear framework for decisions regarding the level of evidence needed to support the modifications that are made to PROMs when they are migrated from paper to electronic devices. Three levels of modification (minor, moderate, and substantial) of the original paper-based PROM to ePROM have been reported, and an effective strategy for testing measurement equivalence (reliability and validity) has been provided [20]. When altering the Mode of Administration (MOA) from paper to tablet, moderate modifications (such as splitting a single item across multiple screens, requiring the patient to use a scroll bar to see all the items or responses) generally require equivalence testing together with usability testing. More recent evidence suggests that previous usability evidence in a representative group is sufficient to assume equivalence, as opposed to per-study testing [37]. Depending on the level of change, additional testing may be required to establish the equivalent reliability and feasibility of ePROMs [34]. If equivalence has been established, then setting up the PROM to correspond to the way it was validated will maintain validity, such as, for example, placing the same questions on each page rather than being split across pages.

Bring your own device (BYOD), where patients use their own device for ePROM administration, seems likely to be the preferred method of administration in the future [37]. There are however technical and practical considerations to take into account. BYOD may reduce costs and allow patients to work on familiar equipment. Things to keep in mind are that patients may turn off in-app notifications, remove the study app, change devices, run out of data or device storage, and be interrupted by other activities on the device [37].

The feasibility and utility of implementing an ePROM for burns has been examined in the US, using the Young Adult Burn Outcome Questionnaire (YABOQ), with real-time benchmarking feedback in a burn outpatient practice [9]. That study examined the data of 12 patients, aged 19–30 years, 1–24 months from injury, who completed the PROM and demographic data on an iPad in the office before outpatient visits. The study found preliminary evidence of the feasibility and potential utility of the real-time use of the burn-specific ePROM. Qualitative results supported the hypotheses that ePROMs can

facilitate communication between patient and provider in burn outpatient settings and help providers identify the clinical issues to address. [9]. These findings have been confirmed in our own work, which has indicated the feasibility of ePROM implementation in burn scar rehabilitation clinics if barriers can be addressed [10,38].

### 3.1.2. Selecting, Administering, and Scoring PROMs

- Selecting

Clinicians may feel it is within their role to select PROMs for implementation but obtaining the input of researchers may be valuable to review the psychometric properties of the appropriate measures. Involving patients in the selection is vital to ensure the content, wording, and layout is appropriate. Choosing an appropriate PROM should take into account the population, the purpose for administration (diagnostic/screening, prognostic, and monitoring), validity, content, recall period, time required for administration, available languages, and ease of use [39]. Questions that clinicians and researchers can ask themselves to guide the selection of PROMs have been added to the resources (Appendix A).

We suggest that clinicians and researchers select burn scar outcomes based on qualitative work that exists in relation to these outcomes, in the absence of consensus regarding the outcomes of importance for people with burn scars. Some of this qualitative work has been included in Appendix A, although it may not be all-encompassing. Qualitative methods of identifying relevant outcomes have been recommended as a precursor to consensus approaches to ensure patient perspectives are incorporated in the final core outcome sets [39].

- Administering

ePROMs can be built into readily available platforms and packages, including Qualtrics, Question Pro, REDcap, Survey Gizmo, and Survey Monkey. Some organisations will have system-wide site licences for these platforms and packages that can be accessed.

During the in-centre or hospital administration of ePROMs, technical and practical issues need to be thought through in advance. This should include determining whether it is possible for the patient to complete the ePROM alone or whether assistance is required and, if a tablet is used, having a secure location where it can be stored (or adding a security cable lock and security code). Making sure instructions are available for each specific ePROM, either verbally or as part of the electronic delivery, should be considered to provide context regarding what the information is being used for and how to complete the ePROMs.

Differences in response options between paper-based and electronic PROMs should be considered when selecting the mode of administration. For electronic measures, it has been suggested that respondents should be able to opt out of answering questions or be able to skip questions [40]. This needs to be considered when applying settings as survey platforms such as Qualtrics require participants to respond to each question before proceeding unless opting out is applied to the forced choice setting.

- Scoring

It is important to consider that implementing electronic score calculations in a web-based platform is a one-time effort and human resources cost. Once calculations are implemented and tested, all future responses on the questionnaires are calculated automatically and are thus instantly available. Scoring questionnaires electronically provides the benefit of scalability and having the results available immediately.

### 3.1.3. ePROM Results, Feedback, Evaluation, and Training

Health professionals may need to consider monitoring ePROM results even when ePROMs are used in research as ePROMs are not neutral activities; they can change the way patients think about their condition [11] and may trigger emotional responses [41]. Visualisation of the results (for example, using graphical displays or visual aids) seems to be valued by patients [42], and using a historical timeline can give the patient and healthcare professional better insight into the evolution of their health status [31,43,44]. Something to

be aware of is that exposure to the ePROM itself can increase the willingness to use it [30], and reviewing the results with a healthcare professional is associated with increased odds of perceiving ePROMs as beneficial [45].

There are many ways in which the ePROM results can be used clinically. For instance, automatic scoring of the PROMs in real time may highlight surprisingly low or high scores for a PROM domain or item and thus indicate the need for open communication and discussion of issues that otherwise may not have been picked up. Identifying patient needs early may contribute to earlier identification of the need for multidisciplinary input. Remote patient follow-up using ePROMs is also possible and may allow more time for care during in-person visits to a burn (after-care) centre. Finally, the ePROM results can be discussed alongside the findings of clinical or physical measurements and may reveal aspects of functioning and disability that give additional insight into the impact of the burn scarring on the patient or the burden of treatment.

To evaluate whether or not the implementation of ePROMs is successful, patient and clinical outcomes, implementation outcomes [46], and mediator/proximal variables (for example, perceived relative advantage of ePROMs) should be considered [47]. Measures that could be considered for evaluation include Patient Experience Measures (for example, satisfaction with training and discussion of ePROM results), and assessing organisational readiness to implement ePROMs [47].

Experts in the field of PROM implementation have recommended training as a critical strategy for PROM implementation [47]. The features of training that should be considered are included in the accompanying resources (Appendix A). However, a recent review of RCTs evaluating the use of PROMs as interventions, which identified training of clinicians regarding PROMs prior to trial commencement, resulted in no obvious impact on the results [48]. Thus, further evidence is needed regarding the effect of training.

### 3.1.4. Overarching Considerations

- Ethical considerations

Health professionals having the capacity and desire to respond to issues that are identified on ePROMs is an important ethical consideration when implementing these measures routinely [11].

- Security

Security requirements for ePROMs will vary depending on the legislation, policy culture, health authority, national policies, and contexts [35]. Early collaboration with Information Technology (IT) personnel is advised in order to overcome organisational IT barriers and obtain support. The storage of information overseas may not be legal in some jurisdictions unless participants are fully informed of the implications and provide written consent. Stored information may need to be non-identifiable in these cases, which may limit the information that can be captured electronically.

- Policy and culture

Broader health service policy and internal culture may influence the implementation and sustainability of ePROM implementation and thus should be considered early. For example, in health services or departments where ePROM implementation is identified as a priority, the resources to support implementation (such as IT support and solutions to overcome privacy concerns) may be readily accessible, or teams may be able to work together to overcome barriers.

- Changing behaviour and readiness for implementing ePROMs

Remarkably consistent barriers to implementing ePROMs have been identified across conditions, settings, and countries, including IT issues, competing demands from existing workloads, and the time and cost of implementation [45]. Our own experience and findings from our case studies support this evidence. The strategies to address these barriers should go a long way towards being ready for implementation, if facilitators of implementation in local settings are identified and considered alongside barriers [47].

- Equity of access

  Equity of access should be considered to ensure that models of care based on ePROMs do not widen already wide disparities in access to healthcare and health outcomes [48]. Making ePROMs available in more than one language is one way of addressing this. If this is not possible, based on lack of cross-cultural validation of PROMs or limited resources to allow purchasing licences in different languages, then the collection of data to inform future equitable access to ePROMs should be considered. This could start as simply as collecting information on the social determinants of health or looking at the acceptability of using ePROMs in people from diverse cultural and linguistic backgrounds [49]. Alternate person-centred methods of PROM administration to elicit patient perspectives and facilitate communication could also assist in capturing the voice of people with health and literacy challenges [50]. These alternate methods could include creating photos or selecting photos from archives [42,50], video-based methods [18,51], digital storytelling [52], and Ecological Momentary Assessments (EMAs) [53]. EMAs are repeated assessments of behaviours and experiences in real-time natural environments, usually administered using technology such as electronic diaries, telephones, and sensors [54].

- Conceptual frameworks

  The use of conceptual frameworks can inform the development of implementation questions and hypotheses in research, the selection of strategies to assist implementation, and the identification of barriers and facilitators to implementation [53]. Recent papers on ePROM implementation [47] and implementation science [53] provide guidance on applying frameworks that may be useful for ePROM implementation, including Normalisation Process Theory [55], the Consolidated Framework for Implementation Research (CFIR) [56], and the Integrated Framework for Promoting Action on Research Implementation in Health Services (I-PARIHS) [57]. In situations when information from ePROMs is fed back to clinicians for use in clinical consultations or to patients, Feedback Intervention Theory may also be applicable in understanding how feedback from ePROMs might work to elicit a therapeutic response (the mechanism of action) [15,58]. The essence of this theory is that feedback can draw a person's attention to gaps between their current and ideal health state, resulting in positive but also sometimes negative effects [15,58].

- Dissemination of information

  To promote equity, dissemination methods that resonate with key stakeholders or people with burn scars should be considered; these could include images or narratives [50]. Involving the target group, for example people with burn scars, in the preparation of information that will be disseminated is a strategy that may assist in ensuring that the messages are appropriately pitched (for example, they should be readable, unambiguous, relevant, and easily comprehended) [59]. Other methods likely to improve the success of dissemination include the use of social media to broaden access to health information [59]. Figure 1 illustrates the processes and strategies that should be considered at different stages of ePROM implementation.

| Prior to implementation | Implementation | Post implementation |
|---|---|---|
| • Obtain necessary support (financial, IT, governance).<br>• Assess organizational readiness for change.<br>• Check functional abilities and mental capacity of your target population (children or adult population) and include an education session on completion of the ePROM for each individual where possible.<br>• Decide mode of delivery – mainframe computer versus tablet/smartphone/access via platform.<br>• Assess workflow and make recommendations for PROMs.<br>• Choose PROMs based on objectives (e.g., goal setting, population, intended use, psychometrics, biopsychosocial, burden).<br>• Assess clinic barriers, enablers, PROM needs (check IP of the ePROM and obtain approvals for use).<br>• Check whether the psychometric equivalence of the ePROM with paper based PROM been established?<br>• Assess IT technicalities.<br>• Engage clinic team to develop plan for overcoming barriers.<br>• Decide whether education is necessary for the target group, clinicians, researchers multi-disciplinary team.<br>• Decide upon ePROM settings (e.g., forced responses, automated reminders).<br>• Check whether ethical and privacy concerns have been considered.<br>• Decide whether assistance will be provided during completion of ePROMs (by who, is it possible, necessary?).<br>• Decide whether summary results will be made available (i.e., graphical result summaries). How will they be provided (electronic, paper-based)? Who will access them (patients, clinicians)?<br>• Check access to appropriate equipment (e.g., printers linked to a tablet so the results can be taken into appointments in real-time).<br>• Consider whether linking with electronic medical records is possible and develop a process to achieve this. | • Ensure assistance is available if a decision was made to offer assistance.<br>• Consider applying a code to data from each patent at the beginning of data capture to make the data non-identifiable or re-identifiable.<br>• Discuss automated graphic results if appropriate<br>• Check access to appropriate equipment each day of administration (e.g., printers linked to a tablet) Consider which staff will approach patients and who can assist patients if all staff are busy.<br>• Check data is removed from the device immediately after it is captured so that no patient or staff member has access to data that they should not see. | • Consider infection control (in centres using tablets).<br>• Think about storage of the data and the device<br>• Evaluate the experience with a satisfaction questionnaire to check feasibility, acceptability.<br>• Link ePROM data with electronic medical records if possible.<br>• Monitor the data captured and referral when symptoms and quality of life are concerning (e.g., alarming signals/symptoms/immediate referral/action).<br>• Periodic evaluation and meetings until PROMs/PREMs become routinized.<br>• Data-informed quality improvement.<br>• (Re)training (if necessary). |

**Figure 1.** Implementation strategy guide across all stages of ePROM implementation.

### 3.2. Case Studies

Our case studies are real-world examples of implementing ePROMs in clinical practice from Europe and Australia. These examples include: (1) a digital pathway based on ePROM data in a community-based scar after-care centre for clinical decision making and (2) implementing ePROMs and graphical displays of information from ePROMs as part of a research initiative to guide the treatment of children with burn scars attending a children's hospital. In Figures 2 and 3, the context, outcome selection, ePROM intervention, and implementation considerations for each case are outlined.

In case study 1 on ePROM implementation in Belgium, the implementation was for research purposes as well as clinical follow-up during post-burn rehabilitation. Prior to the implementation, PROMs were already commonly administered in the setting. A literature review on the advantages and disadvantages of ePROMs [17] and a validation study on the chosen ePROMs [38] were performed prior to implementation. All patients eligible to receive after-care were recorded in the dashboard of the platform used. For every patient, the results of the assessments, ePROM results, and treatments were recorded and grouped per patient and per research study or specific care pathway in the platform. With the onset of the COVID-19 pandemic, the possibility of using their own device or smartphone to complete the ePROMs remotely or on-site was an extra advantage. The digital care pathway, Scarpath [60], allowed the continuation of the research follow-up with ePROMs during periods of restrictions and lockdowns when physical attendance for in-centre visits was postponed or not possible. In the burn scar rehabilitation setting where ePROMs were implemented, our mission was to be patient-centred and holistic, taking into account the biopsychosocial functioning of patients, which the initiative aligned with. When implementing the ePROM pathway, the physical and psychosocial aspects (for example, scar quality and overall quality of life) were bundled into one dashboard

per patient which helped to gain a holistic patient-centred view of functioning at a specific moment in time or over time.

In case study 2 on ePROM implementation in Australia, the implementation was for research purposes initially. Prior to the implementation, PROMs were commonly administered in the burn scar setting in paper form but were not scored in real time. The final results of the research are yet to be published. Preliminary findings identifying the barriers and benefits of implementation indicated that the barriers included: completing PROMs at initial consultations in clinical units, which could be a barrier to natural communication; a lack of capacity by clinicians to respond to some of the issues identified; technology issues; and competing demands [10]. The benefits included use of PROMs that targeted areas of importance to families and relatively safe topics and high value placed on being asked about some topics that would not typically be raised [10].

## 1. Digital care pathway in adults with scars in Belgium

**Context**: The ePROM implementation was performed in Oscare, a multi-disciplinary scar after-care and research centre in Antwerp, Belgium.

**Outcome selection**: The PROMs were selected based on prior validated measures, the WHO's biopsychosocial model of functioning (the ICF framework, created by the World Health Organisation in 2001, and prior research "Meirte J. 2016" indicating that both disease-specific and generic measures as well as the different domains of functioning should be addressed "Meirte et al. 2014".

**ePROM intervention**: Adult patients completed scar specific and generic PROMs (Patient and Observer Scar Assessment Scale and Euroqol 5 dimensions (EQ-5D), Dermatology Life Quality Index (DLQI) quality of life measures) on touchscreen tablet-computers before their first physical therapy appointments. This was combined with objective scar assessments. ePROMs were completed at set time points (baseline, after 1 month, after 3 months, (if applicable 6 months- 1 year)). The ePROMs were filled out at the centre or from home. Digital questionnaires were built using a digital web-based care pathway from Awell Health, a tech company located in Brussels (Belgium). Awell Health developed a Software as a Service (SAAS) in which care pathways were digitalized and patient data was collected electronically via a delivery mode of choice (tablet, computer, smartphone).

**Purpose of ePROM implementation:** The aim for collecting data was both for research purposes as well as for clinical follow-up during rehabilitation. The ePROMs are part of an entire digital care pathway including treatments over time.

### Implementation considerations

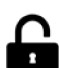 **Security and privacy**: All data was stored in the E.U. with no data transferred to the US or other country. The digital pathway is ISO27001 certified and GDPR compliant, addressing privacy issues. Patient confidentiality was maintained using strict access controls for clinicians and Awell support staff, with data encrypted in transit and at rest and informed consent obtained from patients.

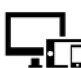 **Delivery method**: Patients who wanted to take part had details entered in the digital pathway and a digital care pathway was created for them. Patients could fill in the questionnaires on an iPad or laptop form the research/after care centre or on their own smartphone. In updates of the digital care pathway a virtual assistant Ava has been added to support the patient throughout the care journey with timely information, assistance with the next steps. The front-end (user-facing part) is designed as a web application that is accessible from the user's browser. Therefore, no additional installation on the user's hardware is required. Every patient starting an after-care program is entered in this digital pathway which allows tracking of intake information, subjective and objective scar assessments, treatment of the patient and automated reminders for follow-up consultations. ePROMS are automatically scored real-time by the pathway and a dashboard of all patients in the after-care process can be accessed by health care workers. Researchers involved use both computers and tablets to look at ePROM results and to visualize the results during consultations to screen for distressing results.

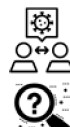 **Infection control**: In between patients the tablet used in the centre was cleaned with antibacterial wipes.

**Findings**: "Meirte et al. 2020" "Digital Health Europe Twinning project Scarpath"

**Figure 2.** Case study of a digital care pathway in adults with scars in Belgium [38,61–64].

## 2. Implementating ePROMs in paediatric scar clinics in Australia

**Context**: Specialist children's hospital outpatient clinics at a major metropolitan burn centre in Australia "Tyack et al. 2021".

**Outcome selection:** PROMs were selected based on previously validated measures for an evaluative purpose (monitoring changes over time), content validity that involved patients themselves in the development, and a focus on health-related quality of life. Cognitive interviewing was conducted to assist in selecting the measures from a pool of potential measures in the study setting.

**ePROM intervention**: Children aged 8 years and older and caregivers of children of aged less than 8 years completed scar specific and generic ePROM health-related quality of life measures (Brisbane Burn Scar Impact Profile, CARe emotional scales –proxy report only, PEDS-QL). Non-English speaking families were not included in the study due to difficulty obtaining ePROMs in other languages and limited resources to purchase questionnaires in languages other than English.

The ePROMs were administered electronically using Qualtrix, a survey software platform. This platform was chosen based on the ability to present graphical summary reports from the ePROMs and include other features such as graduated response circles to obtain answers similar to paper-based PROMs, and open-ended questions. The Qualtrix platform allowed real-time scoring and presentation of graphical displays of the results to clinicians during consultations and comparison to the results of previous consultations.

As the electronic data from the ePROMs could not be entered into the electronic medical records in real-time, the data was printed out and provided to clinicians, who could then send the information to the scanning department along with a medical identifier, for entry into the medical records which could take several days.

**Purpose of ePROM collection/implementation**: Implementation was part of a research initiative in three skin clinics with the intention of informing ongoing routine implementation of ePROMs in clinical practice.

### Implementation considerations

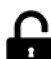 **Security and privacy**: At the beginning of the study it was not clear whether the information stored in the Qualtrics platform might be stored outside of Australia thus participant information and consent forms advised participants that information may be stored outside of Australia. Back-up files of the data collected were stored in a coded form in a password protected secure file on a university server. A list of codes with identifiable data were stored separately from the coded data files. Participant data provided to clinicians were made identifiable by researcher investigators replacing the code used to collect data within Qualtrics with the participant's name. All study communication via text or email was sent out with the clinic, hospital and investigator name clearly displayed.

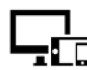 **Delivery method**: All families enrolled in the study were sent an email link to complete the questionnaires electronically (e.g., iPhone, home computer, iPad) prior to their outpatient appointments up to 6-months post-burn (but no more frequently than once a month). When questionnaires were not completed remotely prior to the appointment, iPads or iPhones were used in the study setting immediately prior to appointments. Reminders to complete the ePROMs were sent manually via email or by phone.

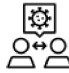 **Infection control**: iPads were wiped over with antiseptic wipes between participants for infection control. However, as the COVID-19 pandemic progressed patients were encouraged to access their iPhones to complete the questionnaires when in the clinic setting to reduce the need for non-clinical contact with patients.

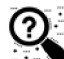 **Findings:** Preliminary results from interviews and field notes involving children with skin conditions (including burn scars) and treating clinicians identified barriers to implementation at the clinic and patient level "Tyack et al 2020".

**Figure 3.** Case study of implementing ePROMs in paediatric scar clinics in Australia (PEDS-ePROM study) [2,10].

### 3.3. Lessons Learned

Based on the lessons learned from our experiences and a review of the evidence, we have 10 recommendations for implementing ePROMs, as illustrated in Figure 4. We suggest that many of these recommendations should be considered prior to full implementation. Cognitive interviews, in which 5 to 10 patients in the target group (patients, parents, or clinicians) complete the PROMs as part of a 'think-aloud' exercise, are ideally conducted to establish the content validity of ePROMs in the study setting. From our own experience, both in clinical settings and in burn scar rehabilitation research, involving the target group in cognitive interviews is highly recommended. Having a stakeholder or advisory committee involved from the start of the initiative can contribute to successful implementation.

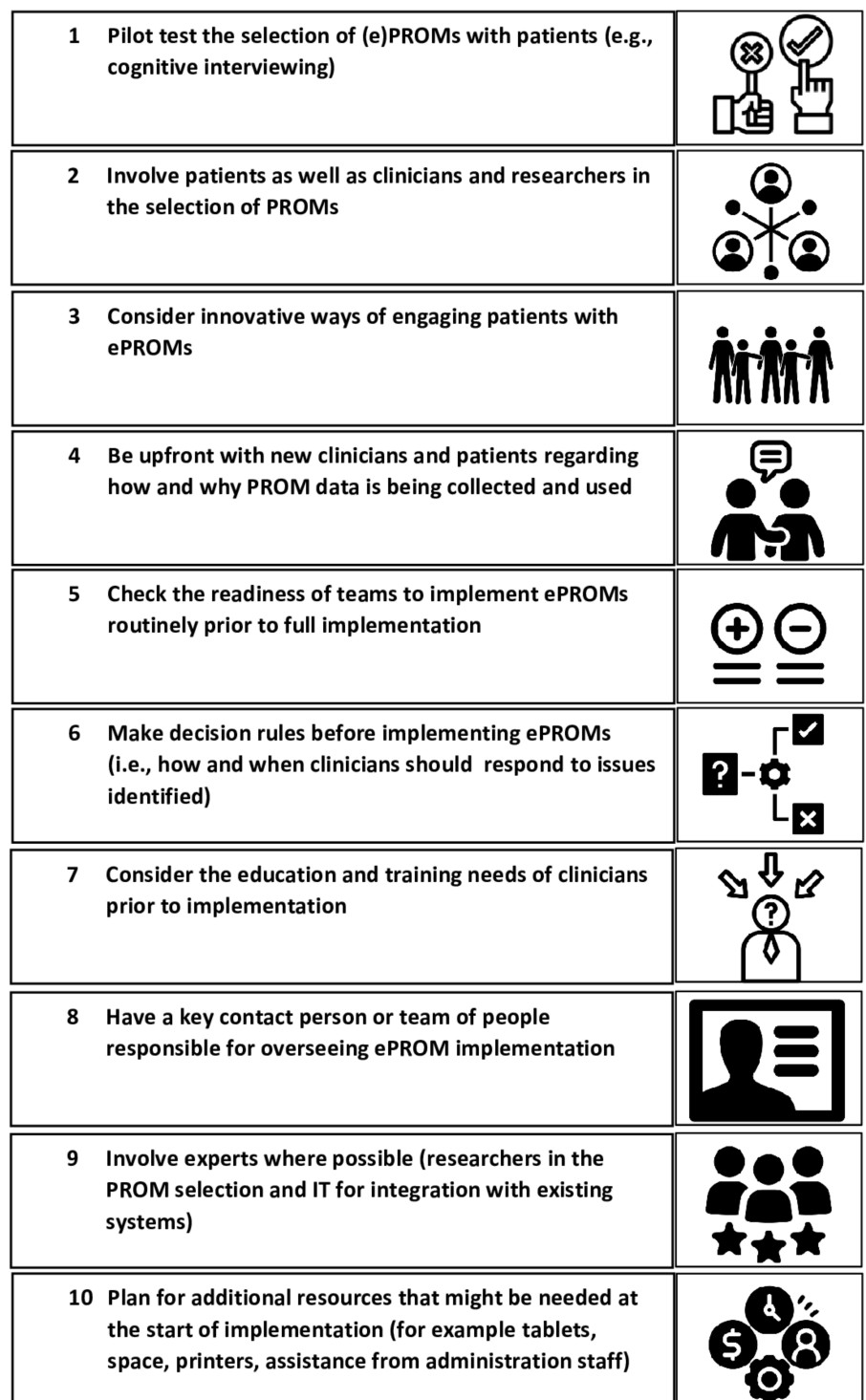

**Figure 4.** Top 10 recommendations for ePROM implementation.

## 4. Discussion

This paper reports on the development of a guide to implement ePROMs in burn scar clinical practice, drawing on evidence, implementation case studies, and the authors' experience. The guide presented should assist researchers and clinicians to consider important processes and strategies when deciding to implement PROMs in clinical practice as part of research, clinical, or quality assurance initiatives. Future work should validate the guide with input from the target group of clinicians conducting scar management as well as by involving patients and families.

In the ideal world, synergy can develop across the stages of burn scar care by capturing the outcomes that matter to patients in each of these stages using ePROMs (and electronic treatment pathways). The first steps in this direction have been made by the Burn Centers Outcome Registry of the Netherlands (BORN), Scarpath Belgium, and the implementation of paediatric ePROMs in Australia (PEDS-ePROM study). For the implementation of ePROMs to be fully realised, understanding the costs of implementing these measures and the impact on long-term outcomes to patients will remain critical to deliver even better value in burn scar rehabilitation [65]. Using feedback from ePROMs implemented in routine burn scar clinical care there is an opportunity to reshape future burn scar rehabilitation.

In the future, the use of the guide and the implementation of ePROMs need to be evaluated for the impact on clinical, patient, health service, and implementation outcomes and tailored for specific contexts. Patient outcomes may include health-related quality of life, communication, adherence to treatment, and reconstructive and psychosocial outcomes. Further work will also be needed to determine how the guide can inform policy [66] and practice, for example in the provision of incentives and support for the use of ePROMs in health services.

Limitations in the development of our guide include a limited evidence base for implementing ePROMs in a burn scar rehabilitation context, involving two countries (Belgium and Australia); thus, broader evidence has been drawn upon. The remarkable consistency of barriers identified across the studies and settings means barriers may be generalisable across conditions and settings, but the facilitators of implementation are likely to be more context-specific [47]; thus, there is a need further examination in relation to implementing ePROMs locally in burn scar rehabilitation settings. This generalisability of the barriers may not extend to low-resource settings as most studies have been conducted in high-resource settings. Work is now needed in low-resource settings to determine the transferability of the processes, strategies, and recommendations in the guide. The potential benefits of ePROMs in these settings include rapid assessment and the use of data to inform assessment, service provision, and public health initiatives [67].

## 5. Conclusions

The guide presented in this paper and the accompanying resources should be viewed as a starting point for implementing ePROMs, with the ultimate goal of enhancing communication with clinical teams and the health-related quality of life of patients with burn scars. As well as addressing the barriers to implementing PROMs that come from the broader literature, researchers and clinicians implementing PROMs should consider factors that impact on implementation in their own setting to tailor the implementation. Refinement of the guide will be required in the future as the ePROM evidence advances in relation to implementing ePROMs in burn scar rehabilitation settings and as more original studies are conducted in multidisciplinary, cross-cultural, and international contexts.

**Author Contributions:** Both authors, Z.T. and J.M., had equal contribution to the conceptualization, methodology, resources, writing—original draft preparation, writing—review and editing, and visualization. All authors have read and agreed to the published version of the manuscript.

**Funding:** This research received no external funding except for the research underlying the Australian case study which was funded by the Children's Hospital Foundation, Australia.

**Institutional Review Board Statement:** Not applicable.

**Informed Consent Statement:** Not applicable except for the case studies of implementing ePROMs. Individual written consent was obtained for participating in research linked to the case studies but as the case studies are presented at the service level or by referring to previously published findings this is not detailed in the case studies.

**Data Availability Statement:** Not applicable.

**Acknowledgments:** We acknowledge the people and patients who contributed to the research and clinical implementation of ePROMs in the case studies, some of whom are listed in the references linked to the in-text citations in the case studies.

**Conflicts of Interest:** The authors declare no conflict of interest. The funders had no role in the design of the study; in the collection, analyses, or interpretation of data; in the writing of the manuscript; or in the decision to publish the results.

## Appendix A. Resources to Support Choosing and Implementing PROMs in Routine Clinical Care

| | Broad Description | Key Information and Relevant Points of Attention |
|---|---|---|
| **Clinical guidelines and processes** | ISOQOL user's guide implementing PROMs in clinical practice. Version 2. 2015 | This guide aims to help clinicians interested in using PROMs in their clinical practice to think through key aspects of implementation. Questions addressed include: 1. What are your goals for collecting PROs in your clinical practice? 2. What resources are available? 3. Which key barriers require attention? 4. Which groups of patients will you assess? 5. How do you select which questionnaire to use? 6. How often should patients complete questionnaires? 7. How will the PROs be administered and scored? 8. What tools are available to assist in interpretating PROMs? 9. When, where, how, and to whom will results be presented? 10. What will be done to respond to issues identified through PROMs? [11] |
| | ISOQOL companion guide and users guide | This guide builds on the user's guide to assist clinicians to address considerations involved in implementing and using PROMs in clinical care using information from real-world case studies [20]. |
| | A PROM implementation cycle across four phases is presented with accompanying instruments, checklists, methods, handbooks, and standards. | The four phases of implementation are: 1. Goal setting 2. Selecting PROs and PROMs 3. Developing and testing quality indicators 4. Implementing and evaluating the PROMs and indicators [68]. |
| **Selecting outcomes and evaluation in burn and burn scar populations** | Patient-Reported Outcomes in routine care—a true innovation but only if used correctly | Choosing the right PROMs depends on the purpose of use, for example whether it is for clinical use or for research, for a single measure or for longitudinal follow-up. This paper suggests three questions to prepare for implementation: 1. Is the goal to measure change at an individual or a group level? 2. Do the PROMs ask questions that are relevant to my patients, clinic, and/or research? 3. Is the PROM validated for my population of interest? [69] |
| | Systematic review of PROMs used in adult burn research in articles from January 2001 to September 2016 * | Thirteen generic PROMs were reported to have evidence of validation data with English speaking adults with burn injuries: 1. Perceived Stigmatization Questionnaire (PSQ) 2. Social Comfort Questionnaire (SCQ) 3. Satisfaction with Appearance Scale (SWAP) 4. Short Form 36-item Medical 5. Outcomes Survey (SF-36) 6. DASH and QuickDash 7. POSAS 8. LLFI-10 9. Community Integration Questionnaire 10. Brief Cope 11. McGill Pain Scale 12. Brief Fatigue Inventory 13. Davidson Trauma Scale Four burn-specific PROMs were reported to have been validated in English with adults with a burn: 1. Burn-Specific Health Scale-Abbreviated (BSHS-A) 2. Burn-Specific Health Scale-Brief (BSHS-B) 3. Young Adults Burns Outcomes Questionnaire (YABOQ) 4. Burn-Specific Pain Anxiety Scale (BSPAS) [3] |

| Broad Description | Key Information and Relevant Points of Attention |
|---|---|
| Systematic review of PROMs used in child and adolescent burn research from January 2001 to March 2013 * | Two generic PROMs were validated in English with children and adolescents with burns:<br>1. Perceived Stigmatisation Questionnaire<br>2. Social Comfort Scale<br>One burn-specific PROM was validated with adolescents with burns:<br>1. Children Burn Outcomes Questionnaire for children aged 5–18<br>Note: the search was limited to articles that had been written up for research purposes [4] |
| Outcomes important to patients with burns during scar management and comparison to burn-specific PROMs based on qualitative research | Eight core outcome domains were identified as important to children and adults in burn scar management:<br>1. scar characteristics and appearance<br>2. movement and function<br>3. scar sensation<br>4. psychological distress, adjustments, and a sense of normality<br>5. body image and confidence<br>6. engagement in activities<br>7. impact on relationships<br>8. treatment burden [70] |
| Seeding the value-based healthcare and standardised measurement of quality of life after burn debate * | Call to use the same quality of life measure at 4 weeks and 3 months post burn. Quality of life measures for the adult burn population established and emerging were: SF-36, EQ-5D, BSHS-B, VR-36 and LIBRE profile. [71]<br>Strong suggestion for teams to pilot a standardised schedule of administration at 4–6 weeks, 3 months, 6 months, 12 months, and 24 months after burn injury date. |
| Core Outcome Set for Burns (COSB) which may have relevance in choosing outcomes to measure preventing or treating patients with burn scars * | Consensus methods used to identify the seven highest ranked outcomes for burns involving clinicians and patients across different stages of burn care:<br>1. death<br>2. specified complications<br>3. ability to do daily tasks<br>4. wound healing<br>5. neuropathic pain and itch<br>6. psychological well-being<br>7. time to return to work/school [72] |
| A systematic review of the quality of burn scar rating scales for clinical and research use and guide for choosing burn scar scales based on an updated systematic review | **Quality of burn rating scales:**<br>• The Patient and Observer Scar Assessment Scale (POSAS)-high quality rating for reliability of the total score and vascularity subscale.<br>• The Vancouver Scar Scale (VSS)-indeterminate ratings for construct validity, reliability, and responsiveness.<br>• The other 17 scales–indeterminant ratings due to methodological issues. [73]<br>**Guide to selecting burn scar rating scales for clinical practice:**<br>The content, purpose, test sample characteristics, and feasibility of eight scales is reviewed in depth, including POSAS, VSS, modified VSS, Matching Assessment using Photographs with Scars (MAPS), and Visual Analogue Scale [74]. |
| Checklists for assessing study quality related to PROM development and validation by COSMIN | Available checklists can be used to assess the methodological quality of PROMs and studies on PROM development or testing. Checklists have been developed by the COSMIN (COnsensus-based Standards for the selection of health Measurement INstruments) group and include:<br>• COSMIN Risk of Bias checklist for PROMs<br>• COSMIN Risk of Bias tool to assess the quality of studies on reliability or measurement error of outcome measurement instruments<br>• COSMIN Study Design checklist [21] |

| | Broad Description | Key Information and Relevant Points of Attention |
|---|---|---|
| **Barriers to PROM implementation** | Remarkably consistent barriers to the implementation of PROMs have been reported across settings and studies and should be considered prior to implementation | Four consistent barriers to PROM implementation have been identified: 1. Technology 2. Stakeholder uncertainty about the use of PROMs 3. Stakeholder concerns about negative impacts of PROM use 4. Competing demands from established clinical workflows [47] Difficulty tailoring to individual patients has also been identified as a barrier to implementation in a review of systematic reviews. [18] |
| **Measurement equivalence of paper and ePROMs** | Recommendations on evidence needed to support measurement equivalence between paper and ePROMs | A general framework for decisions regarding the level of evidence needed to support modifications that are made to PROMs when they are migrated from paper to ePROM devices. Key issues: <ul><li>Determination of the extent of modification required to administer the PROM on an electronic device (minor, moderate, substantial)</li><li>The selection and implementation of an effective strategy for testing the measurement equivalence of the two modes of administration [40].</li></ul> A recent review suggests that previous usability evidence in a representative group is sufficient to assume equivalence, as opposed to per-study testing: Evidence and Recommendations for Clinical Trials and Bring Your Own Device [37]. Consider the nature of each response scale in the instrument to evaluate the appropriateness for migration to the target mode. Response scales such as graduated circles may be difficult to transition from paper to electronic platforms without alteration [75]. |
| **Pediatrics considerations** | Report of the International Society for Pharmacoeconomics and Outcomes Research (ISPOR) and Patient Reported Outcomes (PRO) on good research practices for the assessment of children and adolescents task force | Good research practices outlined are related to: <ul><li>Developmental differences and age-based criteria for PROM administration;</li><li>Content validity of pediatric PROMs (include children as content experts where possible)</li><li>Whether informant-reported outcome instruments are needed</li><li>Appropriate design and format for the target age group</li><li>Cross-cultural issues</li></ul> Recommend that a child's social and developmental contexts be captured in selected paediatric PROMs (e.g., family, school, peer contexts) [17] |
| | Include child self-report perspectives | Child self-report about what makes a good life should be prioritised and may differ to parent report [76–78]. Where it is difficult to obtain self-report, it has been suggested that proxy report be obtained alongside child self-report where possible [78]. |
| | Frequently used generic PROMs | Six PROMs were evaluated to report health states: 1. Pediatric Quality-of-Life inventory 4.0 (PedsQL) 2. Child Health Questionnaire (CHQ) 3. KIDSCREEN 4. KINDL 5. DISABKIDS 6. Child Health and Illness Profile (CHIP). All capture domains of physical, social, emotional health, and school activities. No measure appropriately captured all relevant age-appropriate domains in PROMs: <ul><li>*Parent relations* domain is discussed as vital for child development and well-being. Covered by: CHQ, KIDSCREEN, KINDL (domain family), and CHIP (domain resilience). Not covered by PedsQL.</li><li>*Financial resources* domain measuring the child's perception of their financial resources is covered by KIDSCREEN.</li></ul> The earliest age self-reported PROMs are available is 4 years [79].PROMIS paediatric scales were not considered by Arsiwala et al. (2021) [79]. PROMIS paediatric scales have the advantage of being able to be administered using Computerised Adaptive Testing (CAT's), allowing items to be tailored to individual patients [80]. Considerations to support use of PROMIS scales in paediatric ambulatory care settings have been outlined, including care transitions and privacy. [80] |

| | Broad Description | Key Information and Relevant Points of Attention |
|---|---|---|
| **Additional resources** | Training clinicians to use PROMs in clinical practice | Drawing from case studies of training programs for PROM implementation across three countries and in diverse clinical areas (adult oncology, lung transplant, paediatrics) it was concluded that training sessions should be: <br><br>• Brief <br>• Timed to fit with existing organizational practices <br>• Flexible (group or individual and e-training sessions). <br><br>The most successful element of training was identified as experiential problem-based learning using video/audio-clips and real patient cases. [81] |
| | Best Practices for Migrating Existing Patient-Reported Outcome Instruments to a New Data Collection Mode | Created by an ePRO consortium. Addresses issues that should be considered when migrating existing patient-reported outcome (PRO) instruments to any available data collection mode (e.g., paper, interactive voice response [IVR] system, tablet, web, handheld) [23] |

\* The relevance of burn outcomes to those with burn scars is not clear; thus, we advocate prioritising research targeting burn scar populations to guide the selection of PROMs for people with burn scars until further evidence emerges regarding the relevance of burn outcomes to people with burn scars.

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
