# Peer review of "Electronic Patient-Reported Outcome Measures in Burn Scar Rehabilitation: A Guide to Implementation and Evaluation"

_2673-1991, doi:10.3390/ebj3020025_

Round 1

Reviewer 1 Report

I really enjoyed reading this paper which I think is a very useful introductory guide for considering the implementation of ePROMs.  My comments are very few and minor and intended for the authors to consider whether small amendments would benefit the paper.

Firstly, I wondered if the introduction might benefit from a brief paragraph on the potential benefits of a move to ePROMs (assuming that PROMs are currently in use).  At the moment the introduction justifies the guide on the basis of a shift to digital and e-health generally and not necessarily based on the potential advantages of ePROMs.  Perhaps brief consideration of this would set the scene for why the trials and tribulations of implementation could be worth it?

Methods (lines 84-89).  I understand this is not a systematic review but I wonder if a little more detail regarding how existing evidence, guidance and conceptual frameworks have fed into the guidance might be useful here.  This section is labelled as methods but mainly talks about the organisation of the paper as opposed to methods.

Case studies - whilst findings from the case studies are referenced it would be useful to have a few lines on aims, outputs, and observations from these case studies in this paper.  Whilst interested parties might look up those references and detailed findings, if one purpose of this guide is to enthuse about the move to ePROMs then some burn-specific findings from implementation (brief) could help here.

Recommendations – I think these are broadly sensible but did wonder how much scope there would be to achieve the first two robustly in isolated contexts or services, as opposed to these being more overarching research aims that then inform practice generally.

Appendix 1 – I wasn’t wholly convinced about the utility of this appendix.  It covers quite a few issues and it is not comprehensive.  Is this something to comprehensively develop for the next iteration perhaps?

Minor formatting queries.

Lines 318-320, and 330-333 – Unsure what this text relates to – an error in the draft?

Author Response

Dear reviewer,

Please find enclosed the revision of our paper entitled “Electronic patient-reported outcome measures in burn scar rehabilitation: A guide to implementation and evaluation”.

We thank the reviewer and the editor-in-chief for the valuable comments that have contributed to a significant improvement of the manuscript. Please find enclosed a point-to-point response to the remarks given at the end of this letter. We enclose a copy of the revised manuscript.

We hope that the revised version of the manuscript is now suitable for publication in EBJ.

By resubmitting our manuscript we confirm that all authors details on the revised version are correct, that all authors have agreed to authorship for this manuscript.

Reviewer 2 Report

Thank you for inviting me to review this paper, which was very interesting to read.  The authors offer case studies and practical guidance on the use of e-PROMS in burn care clinics, which I am sure will be very useful to burn care teams. 

My suggestions are mostly minor:

In the section on Population, it could be useful to be clear whether or not this section is referring to work in burns

I could not see that all of the Figures were referred to in the body of the text.

Some of the headings seem inappropriate given the nature of the paper - e.g. 'Results'

Apologies if I missed it, but there are a couple of interesting points in Figure 4 that I didn't see mentioned previously (ie. decisions about responding to issues; cognitive interviewing as part of a pilot) - these are important points and need to be addressed in more detail.

It would also be very useful to give more information about how clinicians then used the data they gained from ePROMs in a clinic setting

re formatting, there are some paragraphs that seem to be an error. Specifically lines 318-320 "Nouns from the Nounproject.com: ‘Online Survey’ by popcornarts, ‘Unlock’ by 318 Nubaia Karim Barsha, ‘computer-tablet-smartphone’ by Lufti Gani Al Achmad, ‘social 319 distance’ by Berkah Icon, ‘Finding solution’ by Prosymbols" 

and lines 330-333 "Nouns from Nounproject.com: ‘Selection’ by Nareerat Jaika, ‘stakeholders’ by Eko 330 Punomo, ‘group’ by Eunji Kang, ‘Explain’ by Adrien Coquet, ‘pros and cons’ by Adrien 331 Coquet ‘rule’ by Kamin Ginkae, ‘customer need’ by Kamin Ginkae, ‘contact card’ by Dan- 332 iel, ‘experts’ by Adrien Coquet, ‘resources’ by Lufti Gani Al Achmad"

The table in Appendix 1 is useful but I think it could be clearer if formatted a little differently - even if only as landscape rather than portrait and not centering the third column would be helpful, I think. 

This sentence needs reviewing and clarifying (line 303) "To promote equity in dissemination methods that resonate with key stakeholders or people with burn scars who experience disparity may be necessary, for example, using images or narratives [47]."

Author Response

Dear reviewer,

Please find enclosed the revision of our paper entitled “Electronic patient-reported outcome measures in burn scar rehabilitation: A guide to implementation and evaluation”.

We thank the reviewers and the editor-in-chief for the valuable comments that have contributed to a significant improvement of the manuscript. Please find enclosed a point-to-point response to the remarks given by you at the end of this letter. We enclose a copy of the revised manuscript.

We hope that the revised version of the manuscript is now suitable for publication in EBJ.

By resubmitting our manuscript we confirm that all authors details on the revised version are correct, that all authors have agreed to authorship for this manuscript.

Reviewer 3 Report

The authors present a paper highlighting recommendations for ePROM implementation in burn scar rehabilitation. Overall, the paper is well written and highlights an area of interest considering the impact of burn scars on burn survivors. I wonder if the recommendations are directly transferable to low resource settings? 

Author Response

Dear reviewer,

Please find enclosed the revision of our paper entitled “Electronic patient-reported outcome measures in burn scar rehabilitation: A guide to implementation and evaluation”.

We thank the reviewers and the editor-in-chief for the valuable comments that have contributed to a significant improvement of the manuscript. Please find enclosed a point-to-point response to the remarks given by the reviewers at the end of this letter. We enclose a copy of the revised manuscript.

We hope that the revised version of the manuscript is now suitable for publication in EBJ.

By resubmitting our manuscript we confirm that all authors details on the revised version are correct, that all authors have agreed to authorship for this manuscript.
